

# Design of load-aware resource allocation for heterogeneous fog computing systems

Syed Rizwan Hassan[1,*], Ateeq Ur Rehman[2,*], Naif Alsharabi[3,4], Salman Arain[1], Asim Quddus[5] and Habib Hamam[6,7,8,9]

[1] Department of Electrical Engineering, Institute of Engineering and Fertilizer Research, Faisalabad, Pakistan
[2] School of Computing, Gachon University, Seongnam, Republic of Korea
[3] College of Computer Science and Engineering, University of Ha'il, Ha'il, Saudi Arabia
[4] College of Engineering and Information Technology, Amran University, Amran, Yemen
[5] Department of Electronics Engineering, University of Chakwal, Chakwal, Pakistan
[6] International Institute of Technology and Management, Commune d'Akanda, Libreville, Gabon
[7] School of Electrical Engineering, Department of Electrical and Electronic Engineering Science, University of Johannesburg, Johannesburg, South Africa
[8] Centre Ville, Bridges for Academic Excellence, Tunis, Tunisia
[9] Faculty of Engineering, Université de Moncton, Moncton, Canada
[*] These authors contributed equally to this work.

Corresponding author
Ateeq Ur Rehman, 202411144@gachon.ac.kr

## ABSTRACT

The execution of delay-aware applications can be effectively handled by various computing paradigms, including the fog computing, edge computing, and cloudlets. Cloud computing offers services in a centralized way through a cloud server. On the contrary, the fog computing paradigm offers services in a dispersed manner providing services and computational facilities near the end devices. Due to the distributed provision of resources by the fog paradigm, this architecture is suitable for large-scale implementation of applications. Furthermore, fog computing offers a reduction in delay and network load as compared to cloud architecture. Resource distribution and load balancing are always important tasks in deploying efficient systems. In this research, we have proposed heuristic-based approach that achieves a reduction in network consumption and delays by efficiently utilizing fog resources according to the load generated by the clusters of edge nodes. The proposed algorithm considers the magnitude of data produced at the edge clusters while allocating the fog resources. The results of the evaluations performed on different scales confirm the efficacy of the proposed approach in achieving optimal performance.

## INTRODUCTION

The adaptation of Internet of Things (IoT) technology provides seamless connectivity of devices under everyday use to the Internet. The IoT devices can sense and transmit information over the Internet and are expected to reach one trillion by 2025 (*Liang, Xing & Hu, 2023*), creating a financial impact of 11% of the world's economy

(*Jarašūniene, Čižiūniene & Čereška, 2023*). For the processing of the massive information produced by these devices, high computational and storage resources are required.

The cloud-centric paradigm consumes resources available in a centralized way to execute the processing tasks related to information detected by the IoT devices. The cloud architecture provides resources for the execution of tasks through cloud servers. The general cloud computing architecture employed for the implementation of applications consists of sensor nodes and cloud servers. The sensor nodes residing at the border of the network consist of sensors for the detection of the environment. The results achieved after the processing of the detected information depend on the sensing frequency and excellence of the information detected by the sensors. In a centralized arrangement, the sensed information is transmitted to the cloud server for further processing. Due to the presence of abundant resources at cloud servers, cloud computing architecture is the most commonly employed architecture for the deployment of applications. The cloud paradigm offers high latency due to its centralized provision of services which restricts the large-scale arrangement of delay-aware applications on such a paradigm.

Fog computing architecture emerges as a paradigm shift that delivers resources in a decentralized manner near the network edge. This provision of resources close to the end devices provides several advantages over centralized provision. The fog computing architecture comprises of multiple layers as depicted in Fig. 1. The sensor layer contains devices that are used for the detection and transmission of information to the fog devices. The resource constraint fog devices residing in the fog layer of the architecture provide rapid execution of information processing tasks by providing services near the end nodes. This provision of computational facilities close to the end devices reduces the latency and network consumption. Moreover, such distributed provision of resources permits the execution of delay-aware applications on a large scale (*Mustafa et al., 2022*). The cloud layer contains the resourceful cloud server to which the information processed at the fog layer is transmitted for further processing. Several applications are implemented using fog computing architecture.

In *Hassan et al. (2020)*, authors have engaged fog computing architecture for the implementation of an e-healthcare system. The authors also evaluated their fog-based proposed approach with the cloud-based employment. For providing smart services to the end-users, the researchers in *Ammad et al. (2020)* presented a multi-tier fog computing approach. The information detected by the edge nodes is processed by the fog nodes by using limited available resources. The size of data produced by an edge node depends on the sensing frequency of the sensor placed on that edge node. For efficient implementation of applications on the fog architecture, fog nodes need to have sufficient computational resources available for processing the information generated by the connected edge nodes. Therefore, resource allocation strategy plays an important part in the efficient implementation of applications on the fog architecture. In this article, we have presented a load-aware resource allocation strategy. The proposed strategy assigns resources to fog nodes according to the load generated by the edge devices connected to them.

The load-aware resource allocation algorithm presented in this research confirms the effective utilization of the available fog resources. Furthermore, the proposed algorithm

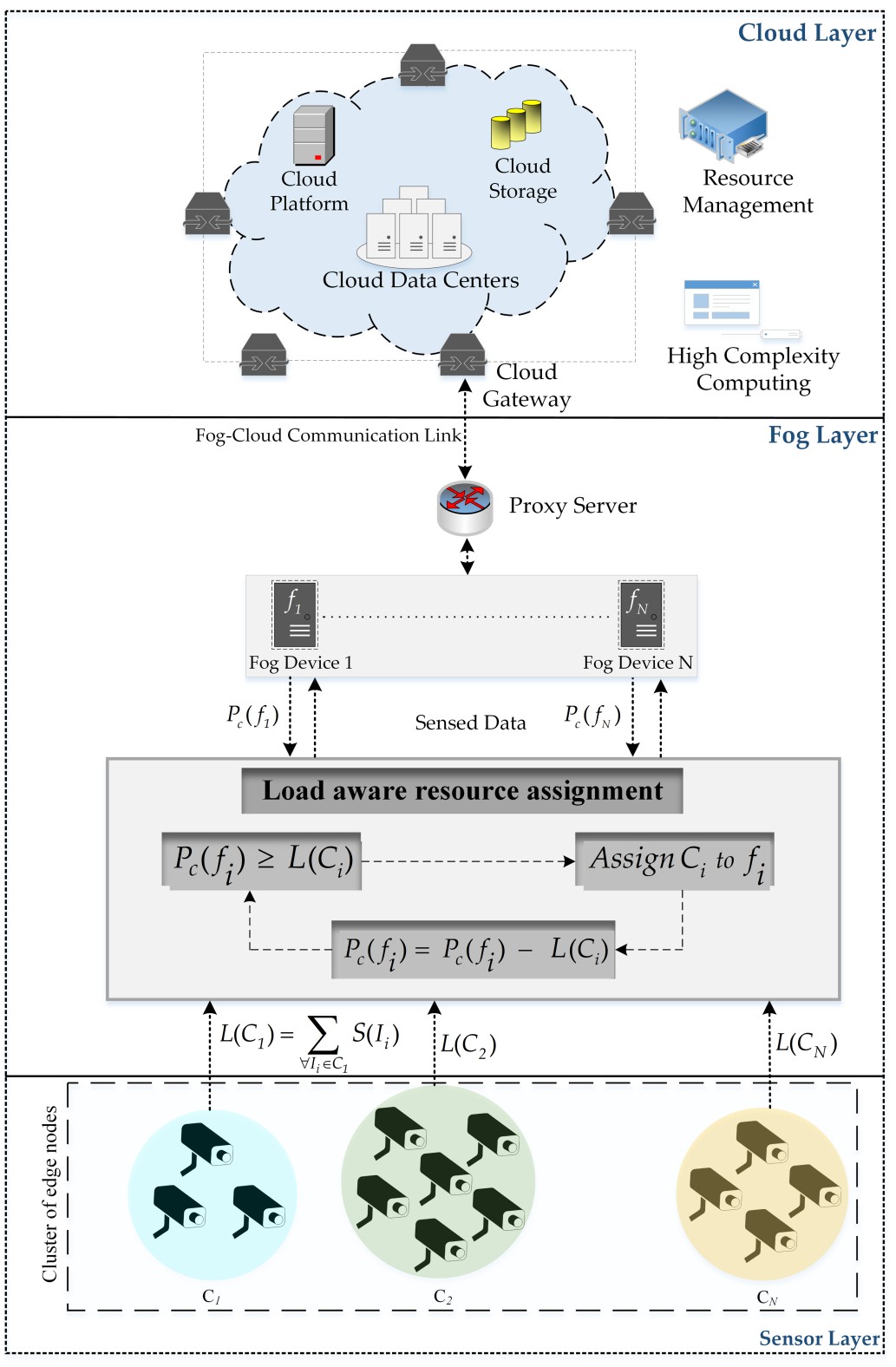

**Figure 1** Fog architecture.

estimates the processing load per fog device and assigns computational resources to the fog nodes accordingly. Several simulations have been executed on multiple scales to validate the effectiveness of the proposed approach as compared to the cloud-based implementation. The network utilization and delay are the parameters observed during all the simulations.

This article is ordered as follows: the subsequent segment delivers a detailed review of literature related to resource and load allocation policies. 'Proposed paradigm and problem formulation' defines the organization of the scenario deployed and the problem statement. 'Proposed Solution' explains the proposed solution for the defined problem. 'Results and Discussion' presents the outcomes of the simulations performed in this investigation and the last segment concludes the research and describes future study directions.

## RELATED WORK

This section reviews the work related to the design policies for efficient implementation of applications on cloud and fog computing paradigms.

The fog paradigm offers services near the edge of the network thus providing a reduction in latency. This delivery of services close to the network edge enhances the end-user experience. However, still, the most important factor in the efficient deployment of applications is the selection of a platform for service deployment. In *Taneja & Davy (2016)*, a resource-aware data analytics platform deployment approach for fog computing networks is presented. The reduction in network cost is achieved by this approach by adaptively deploying the analytics platform. Diverse applications are deployed using a fog computing paradigm that involves heterogeneous devices demanding different amounts of resources to maintain Quality of Service (QoS). A review of different techniques proposed in the literature regarding the maintenance of QoS is presented in *Haghi Kashani, Rahmani & Jafari Navimipour (2020)*. The authors also presented the challenges related to the design and implementation of QoS-aware policies for fog computing environments.

For the implementation of applications in cloud or fog computing environments, several interdependent application modules are executed on different nodes in a network. In *Taneja & Davy (2017)*, the authors presented a module assignment strategy for the Cloud-Fog computing paradigm. This strategy assigns modules to the suitable nodes by keeping in view the resource availability at the node. This is a generic approach and is equally beneficial to be adopted for the diverse applications. Fog devices are distributed throughout the network to provide services on a large scale. Application modules are placed on multiple fog devices to ensure the efficient implementation of applications. A delay-aware module assignment strategy for fog computing networks is presented in *Mahmud, Ramamohanarao & Buyya (2018)*. To meet the rigorous delay necessities of applications, this algorithm assigns the application modules to appropriate fog nodes that guarantee to meet the required service delivery time. The outcomes of the evaluations performed between the proposed and alternate algorithms confirm the effectiveness of the approach.

Fog devices existing in heterogeneous fog computing networks consist of different processing capabilities and have to process diverse kinds of information collected by edge devices. To efficiently execute the application using the fog paradigm, the authors in *Hassan et al. (2022b)* proposed an algorithm that dynamically assigns modules to suitable network

nodes. Moreover, the algorithm takes into account the connection latencies between the edge and fog nodes while assigning the modules. The presented policy is compared with the cloud-based deployment on multiple scales using the iFogSim simulator. The outcomes of the simulations confirm that the proposed algorithm is effective in achieving reduced latency and reduced network utilization. Application placement tasks become a complex managerial problem when dealing with heterogeneous fog environments. The authors in *Al-Tarawneh (2022)* proposed a bi-objective module assignment strategy for heterogeneous fog networks. The algorithm optimally places modules while considering security requirements and criticality levels of application. The non-dominated sorting genetic algorithm II (NSGA-II) is used by the algorithm to solve the formulated bi-objective knapsack problem.

In large-scale networks, the transmission and processing of the massive amount of sensed information generated by the sensor nodes produce high latency and network congestion. Fog architecture is a solution to provide resources in a distributed manner near the sensor nodes. A data duplication placement policy based on the greedy algorithm named MultiCopyStorage is presented in *Huang et al. (2019)*, which provides a reduction in latency. Several simulations are performed using the iFogSim simulator to evaluate the proposed strategy with the CloudStorage strategy, Closest Node strategy, iFogStor strategy, iFogStorZ strategy and iFogStorG strategy. The results of the simulations performed confirm the efficacy of the designed strategy in attaining low latency as compared to the other strategies.

A continuous pain monitoring application based on the cloud computing paradigm is presented in *GJ (2018)* that provides constant supervision of patients in a persistent vegetative state. The proposed application uses the mobile platform for the provision of remote access to information related to patients. A fog computing based remote pain monitoring application is presented in *Shukla et al. (2019)* that senses and processes the pain-related indicators of patients in hospitals. The proposed approach utilizes the available fog resources for the processing of the sensed information. For the provision of remote access to patient information, a web platform is used. The proposed paradigm is compared with the cloud-based implementation and authors performed several evaluations using iFogSim simulator to confirm the effectiveness of the proposed approach. A fog computing based efficient car parking system is presented in *Awaisi et al. (2019)* that provides less delay and network utilization as compared to cloud-based design. An algorithm for the efficient allocation of modules in fog environments containing heterogeneous devices is presented in *Hassan et al. (2022b)*. An algorithm presented in *Taneja & Davy (2017)* presents a job assignment strategy that divides up the modules based on the processing power offered by the system's network devices. The proposed approach efficiently distributes the modules across the fog devices while taking delay, processing capacity, and data size into account. The authors also provide a comparison of the suggested method with conventional systems. *Nandyala & Kim (2016)* present a cloud-based methodology for the delivery of on-demand health facilities. The cloud server acts as a backbone in the proposed design for resource provision and delivery of facilities close to the edge utilizing resources of fog devices. Table 1, compares the proposed strategy to the existing plans on a qualitative level.

**Table 1  Comparison of the proposed approach with existing strategies.**

| Reference | Architecture | Execution cost | Network utilization | Latency |
|---|---|---|---|---|
| *Taneja & Davy (2016)* | Fog-Cloud | Medium | Medium | Medium |
| *Hassan et al. (2022b)* | Fog-Cloud | Low | Low | Medium |
| *GJ (2018)* | Cloud | High | High | High |
| *Shukla et al. (2019)* | Fog-Cloud | Low | Low | Medium |
| *Awaisi et al. (2019)* | Fog-Cloud | Low | Low | Medium |
| *Nandyala & Kim (2016)* | Fog-Cloud | Medium | Low | High |
| Proposed | Fog-Cloud | Low | Low | Minimum |

Integrating wireless power transfer (WPT) into mobile edge computing (MEC) enhances its potential. Meeting the rising demand for intelligent computation offloading in dynamic environments, we focus on real-time, optimal decisions for local or remote computation in wireless fading channels. The authors in *Mustafa et al. (2023)* proposed a binary offloading decision system in a wireless-powered MEC, that utilizes a reinforcement learning-based framework (RLIO) to achieve optimal performance. RLIO boasts an average execution cost below 0.4 ms per channel, enabling real-time and optimal offloading in dynamic, large-scale networks.

In *Atiq et al. (2023)*, the authors introduced Reliable Resource Allocation and Management (R2AM), a fog computing-based framework for efficient resource allocation in IoT transportation, achieving a 10.3% latency reduction and a 21.85% decrease in energy consumption compared to existing strategies. In *Nadeem et al. (2023)*, the authors presented a cloud setting, employing dynamic resource provisioning. *Zaman et al. (2023)* introduced the Deadline-aware Heuristic Algorithm (DHA) for task offloading, considering latency and computing capacity. The DHA significantly reduces total latency (12.67 ms) and offloading failure probability (0.095), outperforming state-of-the-art techniques (19 ms and 0.38 probability).

In large-scale computing systems (LSCSs), the NP-hard problem of load balancing is addressed with the MinMin heuristic. However, MinMin can result in resource imbalance, especially for tasks with lower computational requirements. *Zaman et al. (2019)* proposed a task scheduling heuristic extended high to low load (ExH2LL), a dynamic task scheduling heuristic that outperforms existing algorithms in terms of makespan and resource utilization.

## PROPOSED PARADIGM AND PROBLEM FORMULATION

The fog-cloud computing paradigm offers the benefits of both centralized and distributed computing to the end-users. This paradigm includes a cloud server for the provision of high computational services in a centralized manner and also for distributing the resources near the sensor nodes through fog devices. The fog nodes provide on-demand computational services near the source nodes by using their limited resources (*Fereira et al., 2023*). The fog-cloud computing architecture also provides abundant computational resources through cloud servers for the execution of complex computational tasks. The fog

layer forwards the tasks demanding additional resources to the cloud layer, where these are completed. For the implementation of latency-sensitive applications, this architecture proved to be a better option (*Songhorabadi et al., 2023*) that provides mobility, reduced network utilization and low delay for executing the applications.

Figure 1 presents a three-layer architecture of the fog computing architecture offering distributed resources through fog devices near the edge devices. The cloud server is a resourceful entity existing in the cloud layer of the architecture providing resources for the execution of tasks forwarded by the fog nodes (*Lin et al., 2024*). The fog layer contains fog devices and is responsible for providing a limited number of resources for the execution of the tasks. The sensor layer consists of devices capable of detecting and transmission of sensed information over the Internet. Each device in the system is accountable for the implementation of some application modules (*Hassan et al., 2020*). Different amounts of resources are available at fog devices existing in heterogeneous fog computing environments (*Sharifi, Hessabi & Rasaii, 2022*). These fog nodes provide a reduction in latency by processing information near the sensor nodes. Fog nodes also reduce the processing load on the cloud server by providing preliminary processing using their limited resources.

The heterogeneous nodes present in the network are comprised of dissimilar processing capabilities. The random access memory (RAM) and central processing unit (CPU) are the parameters defining the processing capability of a fog node. The processing capability of the $i$th fog node ($f_i$) is defined as:

$$P_c(f_i) = < CPU_i, RAM_i > . \tag{1}$$

There are total $M$ fog devices present in the architecture and fog layer processing capability is the totality of separate capabilities of all the fog devices present in the layer and expressed as:

$$N = \sum_{i=1}^{M} P_c \{f_i\} . \tag{2}$$

The end devices residing at the edge of the network transmit the sensed information to the parent fog devices. The fog devices process the received information using their limited resources and transmit useful information to the end-users. The tasks demanding more resources than those available at the fog nodes are transmitted to the cloud server by the fog nodes for execution. The IoT devices are available at the user end for the detection and transmission of sensed information. Different types of sensors are connected to these edge devices. The information collected and transmitted by these end devices depends upon the type of sensor. The size of data detected by an end device depends on the frequency of the sensor involved. Different types and sizes of data are generated by the edge nodes in heterogeneous fog networks.

In general fog-cloud computing networks, a group of edge devices connected to a fog device is termed as a cluster of edge nodes. The set $C_i$ denotes the cluster of edge devices connected to $f_i$. The total number of edge devices present in the system is defined by a set $K$

$(K = I_1, I_2, I_3 \dots I_k)$. Different types of edge devices with dissimilar sensing rates are present in a cluster. The sensing frequency of the $i_{th}$ end node $I_i$ is represented as $R_i$. The size of information detected by an edge node depends on the frequency of the sensor devoted to that edge device. The volume of information generated by the $i_{th}$ edge device $I_i$ is denoted by $S(I_i)$ and is proportional to the sensing rate $(R_i)$ of the sensor attached to the edge device. The edge devices with different types and volumes of sensed information are part of a cluster attached to a fog node. So, the volume of information generated by a cluster is the sum of the individual sensed volume generated by each edge device present in a cluster which can be calculated by:

$$L(C_i) = \sum_{\forall I_i \in C_i} S(I_i). \tag{3}$$

The resources available at a fog node are used to process the information received from the attached cluster of sensor nodes. Fog nodes are resource-constrained devices so the tasks demanding additional resources are shifted to the resourceful cloud server for execution. So, to efficiently deploy applications on the fog computing paradigm, resource distribution according to the information to be processed is very important. The optimal assignment of the clusters of edge nodes to suitable fog nodes plays an important part in the optimum utilization of the fog resources.

In the conventional placement of applications on the fog architecture, the size of information sensed by the edge clusters is not considered during the allocation of fog resources which results in high utilization of resources and produces additional delay. In such circumstances, the computational load received by the fog nodes from edge devices is irrespective of the resources available at the fog nodes. When a fog node is linked with a cluster of edge devices producing a high amount of data, the resource constraint fog node transfers most of the tasks to the cloud server for execution. This concurrent connection to the cloud server introduces an additional delay in the execution of tasks. Moreover, this oblivious distribution of fog resources produces high network consumption. For efficient deployment of latency-sensitive applications on the fog-cloud computing paradigm, we have proposed a resource-aware task assignment algorithm. The proposed algorithm offers optimum resource utilization and reduces latency by connecting appropriate fog devices to edge clusters.

For evaluating the proposed strategy, the simulations are performed using the iFogSim toolkit. The distributed camera-based application for surveillance using (*Awaisi et al., 2021*) is redefined and executed on multiple scales in this research to evaluate the proposed strategy. The Distributed Data Flow model (DDF) is used in this study for a better understanding of the distributed computing components (*Fang et al., 2021*).

The directed acyclic graph (DAG) of the deployed application is shown in Fig. 2. The vertices in this model represent the application modules and data flow between modules is represented by the edges. The inclusion of a DAG is vital for orchestrating the complex interplay of tasks within the system. This DAG captures the order in which the tasks like image processing and motion detection, should occur, ensuring a logical sequence. For instance, it helps distribute tasks efficiently across devices by considering their specific

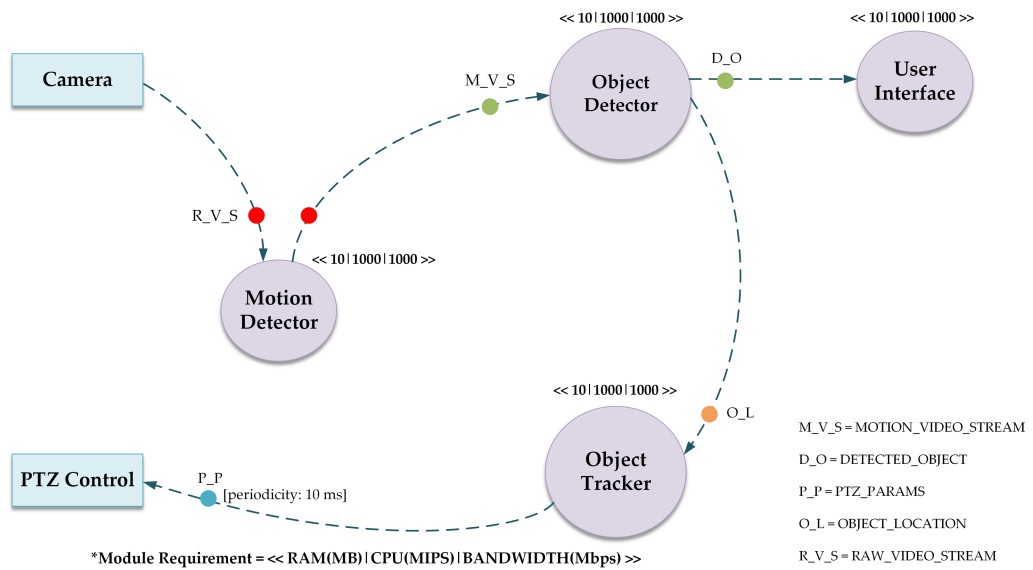

**Figure 2** Directed acyclic graph of the deployed application.

capabilities, and optimizing the use of computational resources in a fog computing environment. The DAG facilitates dynamic task allocation, particularly important in surveillance scenarios where workloads can vary. In essence, the DAG is a crucial tool for designing an effective, scalable, and adaptable distributed surveillance system in the simulated iFogSim environment, optimizing task execution and resource management.

The DAG of the application contains several modules, namely Motion Detector, Object Detector, Object Tracker, PTZ Control and User Interface. Cameras are attached to the Motion Detection module to record video feeds for object motion detection. The Motion Detection module sends this information to the Object Detector module when motion is detected. The Object Detector module then locates the item and calculates its coordinates. In order to effectively track the identified item, the Object Tracker module computes the PTZ configurations for the camera using the data collected by the Object Detector module. After receiving the PTZ configuration, the PTZ Control module adjusts the cameras appropriately. Lastly, the User Interface module delivers filtered video streams gathered from the Object Detector module to the user's device for improved viewing of the monitored item.

The basic unit of communication between the modules is termed as tuple. The tuple is of different lengths encapsulating the data to be processed and information regarding resources required for such processing. The circles of different colors are used in the DAG model for describing the mapping of tuples. For example, the type of tuple M_V_S is released by the Motion Detector module on the reception of a tuple of type R_V_S.

Table 2 below explains the symbols and formulas used in equations and algorithms, making it easier for readers.

## PROPOSED SOLUTION

In this research, we have proposed a load-aware resource assignment model that efficiently accomplishes the links between the edge node clusters and the fog nodes. The proposed strategy calculates the size of sensed data by the edge clusters. Subsequently, the approach assigns appropriate fog resources to the clusters by assigning suitable fog nodes having sufficient resources to process the cluster-generated information. The size of data to be processed generated by the clusters is calculated by accumulating the size of information produced by individual edge nodes contained within the cluster. The proposed algorithm is given below.

---

**Algorithm 1** Algorithm: Load-aware resource assignment algorithm for fog-cloud paradigm

---

Input: Fog devices $f_i \in Layer\,2$, Clusters of edge devices $C_i \in Layer\,3$
Output: Allocation of suitable edge devices to fog nodes
1:  $AC \leftarrow \{\}$;
2:  $UC \leftarrow \{C_1, C_2, C_3 \ldots C_N\}$;
3:  $CS \leftarrow \{\}$;
4:  for each $C_i \in UC$ do
5:    for each $f_i$ do
6:      if $(P_c(f_i) \geq L(C_i))$
7:        Assign $C_i$ to $f_i$;
8:        $P_c(f_i) = P_c(f_i) - L(C_i)$;
9:        $AC \leftarrow C_i$;
10:       end
11:    end for
12:  end for
13:  for each $C_i \in UC$ do
14:    if $(C_i \notin AC)$
15:      $CS \leftarrow \{C_i\}$;
16:      $AC \leftarrow C_i$;
17:    end
18:  end for

---

## RESULTS AND DISCUSSION

For evaluating the proposed design, several scenarios are created on multiple scales using the iFogSim Simulator. The distributed camera-based monitoring application is deployed using the proposed approach. In all the simulation scenarios, eight areas are under observation. Each area is monitored while using a single fog node. All the fog nodes are linked to the cloud server through a proxy server. The cameras are connected to fog nodes to consume fog resources for the processing of the information stream captured by these cameras. The cameras attached per fog node are varied in each setup for the evaluation of

Table 2 Network metrics used in evaluations.

| Symbol | Definition |
|---|---|
| $f_i$ | $i_{th}$ fog node. |
| $P_c(f_i) = \; <CPU_i, RAM_i>$ | Processing capability of the $i_{th}$ fog node. |
| $M$ | Total fog devices present in the architecture. |
| $N = \sum_{i=1}^{M} P_c \{f_i\}$ | Processing capability of fog layer. |
| $C_i$ | The cluster of edge devices connected to $f_i$. |
| $K = I_1, I_2, I_3 \dots I_k$ | Set defining total number of edge devices present in the system. |
| $I_i$ | $i_{th}$ end node. |
| $R_i$ | Sensing frequency of $I_i$. |
| $S(I_i)$ | The volume of information generated by the $I_i$. |
| $L(C_i) = \sum_{\forall I_i \in C_i} S(I_i)$ | The volume of information generated by a $C_i$. |

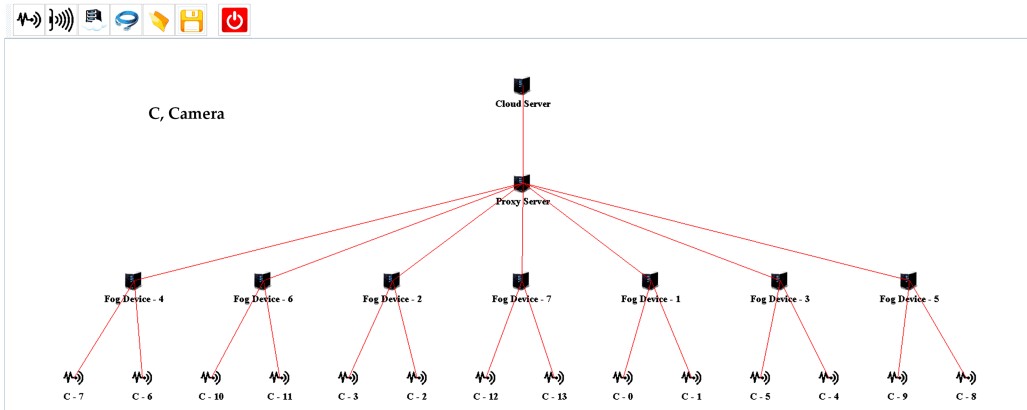

Figure 3 IFogSim topology of the fog-based implementation.

the proposed strategy. Two cameras are initially attached per fog device which is increased in each succeeding scenario. The sensors installed in our evaluations are according to the strategy of *Sharifi, Hessabi & Rasaii (2022)*. The simulation model of one of each setup created in iFogSim for the evaluation of the fog paradigm and cloud computing paradigm is shown in Figs. 3 and 4 respectively. Initially, the cluster attached to the fog device consists of three cameras. The size of the cluster is increased in each subsequent scenario to evaluate the proposed algorithm. The parameters observed during all the assessments are latency, network utilization and cost of execution at the cloud. Tables 3 and 4 describe different network configurations and parameters used in our simulations. The sensor rate of the sensors installed in our simulations is between 5 ms to 20 ms. The simulations are executed on multiple scales for evaluating our proposed cluster assignment strategy with the traditional cloud and fog computing deployments.

Network utilization, execution cost and delay are the metrics observed during the evaluation of the proposed policy with the conventional paradigms. A comparison of

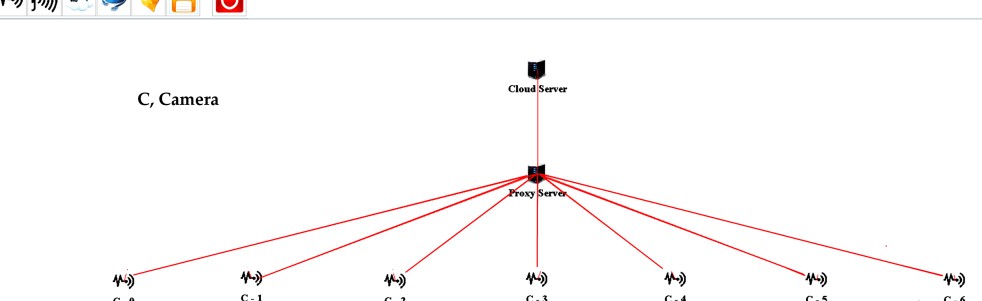

**Figure 4  One of the setups created for simulating the cloud-based approach.**

**Table 3  Types of tuples.**

| Parameter | Cloud | Proxy | Fog device | Edge node |
|---|---|---|---|---|
| Level | 0 | 1 | 2 | 3 |
| Rate per MIPS | 0.01 | 0 | 0 | 0 |
| Random access memory (GB) | 40 | 4 | 2–4 | 1 |
| Downlink bandwidth (MB) | 10,000 | 10,000 | 10,000 | 10,000 |
| CPU power (MIPS) | 44,800 | 2,800 | 2,000–4,000 | 500 |
| Uplink bandwidth (MB) | 100 | 10,000 | 10,000 | 10,000 |

**Table 4  Types of tuples.**

| Tuple type | Tuple CPU length (MIPS) | Network length |
|---|---|---|
| CAMERA | 1,000 | 20,000 |
| M_V_S | 2,000 | 2,000 |
| D_O | 500 | 2,000 |
| O_L | 1,000 | 100 |
| P_P | 100 | 28 |

network utilization during the implementation of an intelligent surveillance application on multiple scales using the cloud, fog and proposed design is presented in Fig. 5. The proposed policy provides a significant reduction in network consumption by assigning appropriate fog resources to clusters of cameras. The size of the information sensed by the edge nodes is calculated using the sensing rate of the sensors (*Hassan et al., 2022a*). The proposed algorithm assigns a cluster of edge devices to a suitable fog node to get the optimum performance of the paradigm. This optimum allocation of fog resources to edge nodes according to the size of information to be processed successfully minimizes the total network consumption. The network utilization observed in the traditional fog architecture is due to the disproportionate provision of fog resources. In the cloud architecture, all the detected information is handled by the cloud server resulting in high network consumption.

A significant reduction in delay is realized by using the proposed algorithm as compared to the cloud and fog paradigm as shown in Fig. 6. The cloud paradigm offers resources in
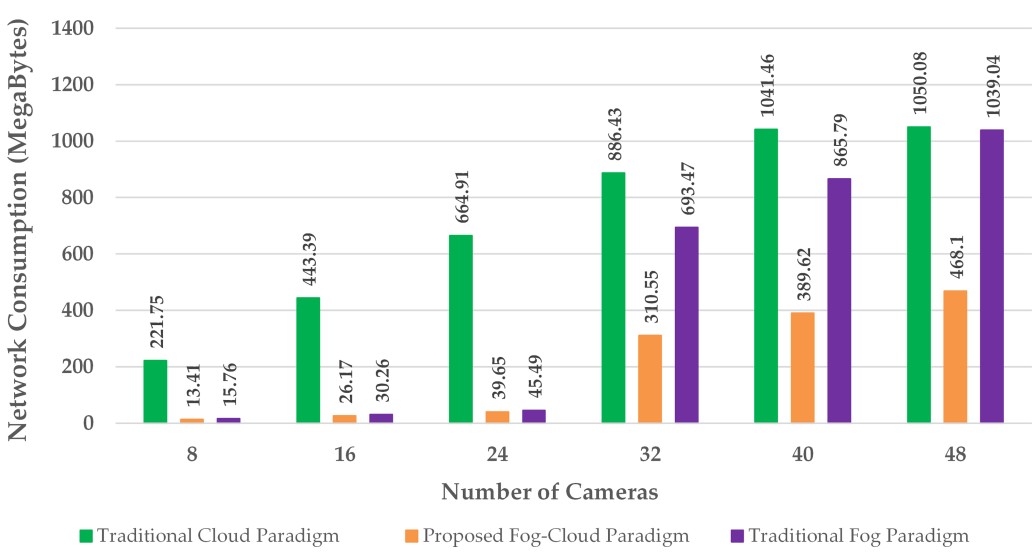

**Figure 5  Comparison of network utilization.**

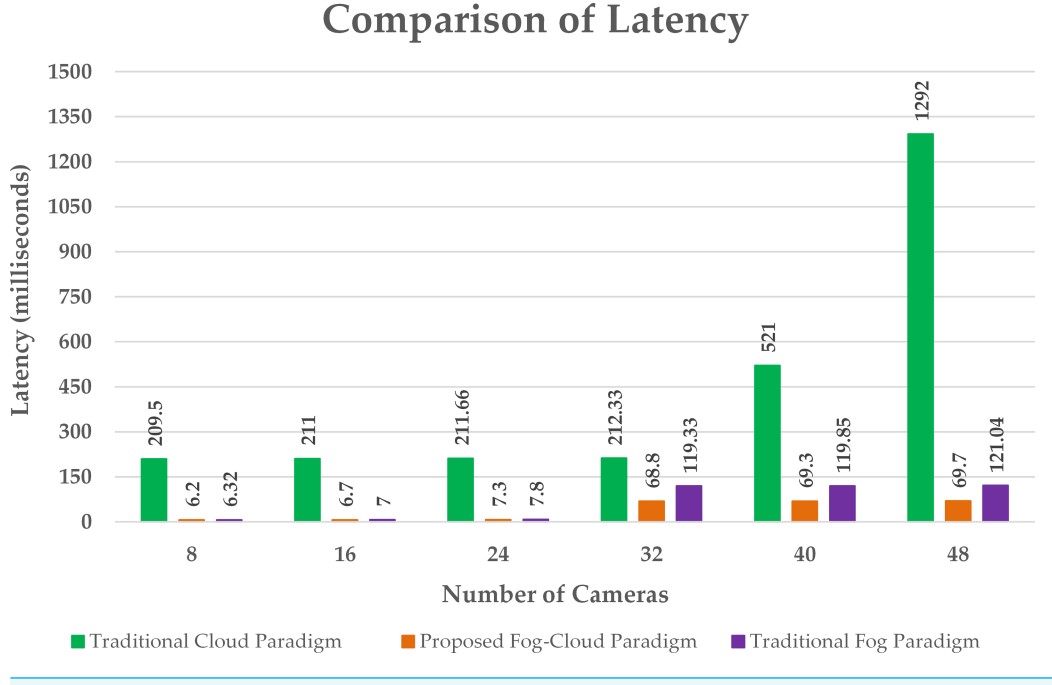

**Figure 6  Latency comparison.**

a centralized manner producing high communication delay. The delay in the cloud-based paradigm rises with a rise in the number of edge devices (*Liu et al., 2022*). However, fog computing provides resources near the sensor nodes for the processing of the sensed information. This provision of computational resources adjacent to edge nodes reduces

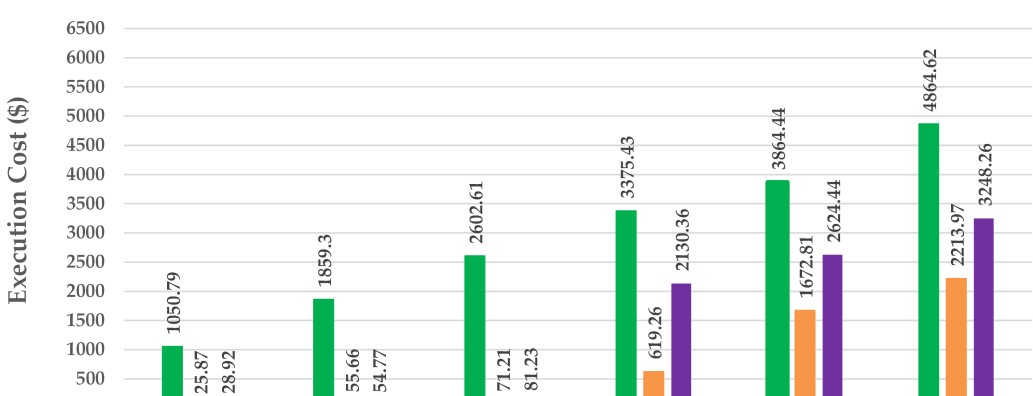

**Figure 7** Execution cost at cloud.

the latency. The proposed algorithm provides resources rendering to the requirement of the edge nodes which further reduces the offered delay. The designed algorithm cuts the processing load to be handled by the cloud server by offering appropriate fog resources which minimizes the execution cost at the cloud as shown in Fig. 7. This allocation of suitable fog resources according to the demand of edge clusters is not obtainable in existing fog deployments.

## CONCLUSIONS

In this study, we developed a resource allocation strategy for fog nodes that efficiently caters to the information size generated by connected end nodes. This tailored scheme optimizes fog resource utilization, ensuring smooth task completion at fog nodes. Through calculating the size of sensed data from clusters of edge nodes and intelligently assigning fog nodes to end device clusters, our algorithm demonstrates a sophisticated approach. The resulting optimal allocation and management of fog resources lead to a significant reduction in network consumption and latency. Simulations on multiple scales using the iFogSim toolkit validate the efficacy of our designed approach in diminishing latency, network utilization, and processing costs at the cloud.

However, it is crucial to recognize certain limitations within our proposed algorithm. Notably, the current framework lacks the capability to address node failure issues, presenting an area for future improvement. Furthermore, the reliance on the sensing rate factor as the primary parameter for resource allocation suggests a need for a more nuanced approach. To enhance the algorithm's robustness, future iterations must consider additional parameters in the allocation process.

Looking forward, our future work entails the deployment of multiple applications using the established approach. A strategic focus will be placed on addressing the identified

limitations, particularly the integration of an artificial intelligence-based framework into the resource allocation process. This forward-thinking approach aims to not only overcome current constraints but also to advance the adaptability and effectiveness of our strategy within fog-cloud computing paradigms. This continuous refinement reflects our commitment to pushing the boundaries of research and innovation in this field.

### Funding
The Natural Sciences and Engineering Research Council of Canada (NSERC) and New Brunswick Innovation Foundation (NBIF) financially supported this global project. The funders had no role in study design, data collection and analysis, decision to publish, or preparation of the manuscript.

### Grant Disclosures
The following grant information was disclosed by the authors:
The Natural Sciences and Engineering Research Council of Canada (NSERC).
New Brunswick Innovation Foundation (NBIF).

### Competing Interests
The authors declare there are no competing interests.
 Habib Hamam is the founder of Bridges for Academic Excellence, Tunis, Centre Ville, Tunisia.

### Author Contributions
- Syed Rizwan Hassan conceived and designed the experiments, performed the experiments, analyzed the data, performed the computation work, prepared figures and/or tables, and approved the final draft.
- Ateeq Ur Rehman conceived and designed the experiments, performed the computation work, authored or reviewed drafts of the article, and approved the final draft.
- Naif Alsharabi performed the experiments, prepared figures and/or tables, authored or reviewed drafts of the article, and approved the final draft.
- Salman Arain analyzed the data, authored or reviewed drafts of the article, and approved the final draft.
- Asim Quddus performed the computation work, authored or reviewed drafts of the article, and approved the final draft.
- Habib Hamam conceived and designed the experiments, prepared figures and/or tables, authored or reviewed drafts of the article, and approved the final draft.

### Data Availability
 The raw data and code are available in the Supplemental Files.

## Supplemental Information

Supplemental information for this article can be found online at http://dx.doi.org/10.7717/peerj-cs.1986#supplemental-information.

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
