# Peer review of "Design of load-aware resource allocation for heterogeneous fog computing systems"

_PeerJ Computer Science, doi:10.7717/peerj-cs.1986_

## Round 0.1 · original submission · Major Revisions

The authors should revise the article based on the comments of the reviewers and resubmit with detailed response.

Reviewer 2 has suggested that you cite specific references. You are welcome to add it/them if you believe they are relevant. However, you are not required to include these citations, and if you do not include them, this will not influence my decision.

**Language Note:** The review process has identified that the English language must be improved. PeerJ can provide language editing services - please contact us at [email protected] for pricing (be sure to provide your manuscript number and title). Alternatively, you should make your own arrangements to improve the language quality and provide details in your response letter. – PeerJ Staff

Reviewer 1 ·

Basic reporting

Clarity of the article is sound. References need significant improvement. Paper structure is fine. Figures need revision.

Experimental design

The article fit in the aims and scope of journal. Research gap is well identified and relevant. Methods need improvement.

Validity of the findings

Findings are valid, however some more experiment are suggested to support the argument. Conclusion can be improved by enhancing the discussion about model limitations and future work.

Additional comments

Article Strengths:
The topic is relevant. The problem is important and well-motivated. The proposed approach to
solution is innovative. The experiments are convincing and well discussed. However, some major
improvements are required, as given below.
Suggestions:
 A table of Notations and Definitions should be added that explains all symbols and
formulas used in equations and algorithms.
 Some sentences are too long. It is recommended to break them to enhance the
 The authors simulated three paradigms, cloud, FOG and Mixed Cloud-FOG as their
proposed model. However, for the proof of concept these simulations along with the
proposed technique should be compared some baseline and state of the art for better
understanding of results.
 Add more simulation details. Such as the environment settings, number of nodes and
controllers, data transmission rate, Number of tasks mapped to MIPS etc.
 There are three performance parameters on the basis of which results are generated. The
mathematical foundations of your (DAG, Latency and Execution) modeling is week in
the proposed methodology. See some relevant papers (an example is given below for
reference) and adopt the benchmark models for your work.
o A load balanced task scheduling heuristic for large-scale computing systems.
2019
 Overall the visual quality of figures should be enhanced without blurriness.
 What are the limitations of your proposal?.Pls enhance the discussion of future work in
conclusion section.
 References have issues. Pls update them. Out of 24 references the 10 references are more
than 5 years old. And 4-5 conferences are cited. Try to get their journal published
versions and replace accordingly. Further, add 5-6 recent articles that are closely related
to the manuscript in terms of Resource Management, load balancing and FOG/Cloud
Computing. For example;
o Reliable Resource Allocation and Management for IoT Transportation Using Fog
Computing. 2023
o Runtime Management of Service Level Agreements through Proactive Resource
Provisioning for a Cloud Environment. 2023
o Deadline-aware heuristics for reliability optimization in ubiquitous mobile edge
computing. 2023

Annotated reviews are not available for download in order to protect the identity of reviewers who chose to remain anonymous.
Cite this review as

Reviewer 2 ·

Basic reporting

The authors provided a scheme for a reduction in network consumption and delays by eûciently allocating fog resources to clusters of edge nodes. The idea is valid but the paper needs following major improvements.

In the abstract, the authors state that "The most suitable candidate among the available computing paradigms for the execution of delay-aware applications is the fog computing paradigm". I do not agree with this solid statement as we have other optimal approaches such as edge and cloudlets. This solid statement should be revised.
The second part of the abstract points out the proposed scheme but does not provide internal details, such as approach eg. are author proposing a heuristic method, numerical optimization, or some AI-based algorithm.

The introduction and related work is supported with old references. There is a need to support this section with recent works: Please go through the following papers.
https://link.springer.com/article/10.1007/s10586-021-03376-3
https://ieeexplore.ieee.org/document/9850011
https://link.springer.com/article/10.1007/s10586-022-03700-5

Experimental design

What is the need for the acyclic graph in the methodology? There is a need os detailed discussion in this aspect.
The authors do not provide the internal settings used in IFog simulator.
A high-level diagram is suggested presenting the complete system model.

Validity of the findings

The results seem to be valid. No need to modify this section.

Cite this review as

---

## Round 0.2 · accepted · Accept

The article is accepted as per revisions and reviewer comments.

Reviewer 1 ·

Basic reporting

English language of the article is improved.

Experimental design

The experimental designs are well structured and portray the better understanding.

Validity of the findings

Result findings are valid and concrete with reasonable arguments.

Cite this review as

Reviewer 2 ·

Basic reporting

The authors have addressed all previous concerns listed in the first revision. Now the article is accepted.

Experimental design

No commnets

Validity of the findings

No comments

Cite this review as